# Antidepressants Target the ST3GAL5–GM3 Lipid Pathway to Suppress Microglial Inflammation

**DOI:** 10.3390/ijms26199733

**Published:** 2025-10-07

**Authors:** Gaku Hayasaki, Hiroto Izumi, Yasuo Morimoto, Reiji Yoshimura

**Affiliations:** 1Department of Psychiatry, University of Occupational and Environmental Health, Kitakyushu 807-8555, Japan; gaku-hayasaki@med.uoeh-u.ac.jp; 2Center for Stress-related Disease Control and Prevention, University of Occupational and Environmental Health, Kitakyushu 807-8555, Japan; h-izumi@med.uoeh-u.ac.jp (H.I.); yasuom@med.uoeh-u.ac.jp (Y.M.); 3Department of Occupational Pneumology, Institute of Industrial Ecological Sciences, University of Occupational and Environmental Health, Kitakyushu 807-8555, Japan

**Keywords:** venlafaxine, vortioxetine, microglia, ST3GAL5, GM3 ganglioside, cytokine signaling, antidepressants, inflammation, major depression

## Abstract

Major depression (MD) is associated with chronic inflammation and impaired neuroplasticity; however, the cellular mechanisms underlying antidepressant action remain incompletely understood. We performed transcriptomic profiling and functional validation in human microglia treated with venlafaxine (VEN) and vortioxetine (VOR), or with stable ST3GAL5 overexpression (ST3GAL5OE). Differential expression analysis, enrichment studies, and functional assays using NF-κB-RE-NlucP and SIE-NlucP reporter lines were conducted to assess the impact on inflammatory signaling. Microarray analysis identified 41 genes consistently upregulated and 316 consistently downregulated across VEN, VOR, and ST3GAL5OE conditions. Upregulated genes were enriched for synaptic organization, whereas downregulated genes were associated with nitric oxide biosynthesis and pro-inflammatory pathways, including Rap1, MAPK, and PI3K-Akt signaling. Functional assays confirmed that VEN and VOR suppressed cytokine-induced NF-κB and STAT3 activation, effects that were recapitulated by exogenous GM3 treatment and ST3GAL5 overexpression. Chronic exposure to VEN or VOR produced more modest, pathway-specific suppression, supporting convergence on the ST3GAL5–GM3 axis. These findings extend the conventional monoaminergic model of antidepressant action by highlighting the ST3GAL5–GM3 lipid remodeling axis as a novel regulatory pathway that attenuates microglial inflammatory signaling. Although validation in primary microglia and in vivo models is required, our results suggest that this axis could serve as both a therapeutic target and a candidate biomarker for inflammation-associated MD.

## 1. Introduction

Major depression (MD) is among the most prevalent chronic psychiatric disorders and a leading cause of disability worldwide, affecting more than 300 million people [1]. Major depression (MD) is characterized by persistent low mood, anhedonia, cognitive dysfunction, and somatic symptoms, often resulting in long-term functional impairment. Selective serotonin reuptake inhibitors (SSRIs) and serotonin–noradrenaline reuptake inhibitors (SNRIs) remain first-line pharmacotherapies. These agents have demonstrated substantial efficacy in alleviating depressive symptoms for many patients, supporting their continued role as standard treatments [2].

The classical monoamine hypothesis—attributing depression to deficits in serotonin and noradrenaline—has guided decades of drug development. However, this framework alone does not fully account for the delayed onset and only moderate response rates of conventional antidepressants [3,4,5]. Contemporary models therefore consider MD as a multifactorial disorder in which neuroplasticity, stress physiology, metabolic state, and neuroimmune signaling interact to shape risk and treatment response [6,7,8,9]. Within this broader view, inflammation is one of several modulatory dimensions rather than a singular cause.

Microglia (MG), the resident immune cells of the central nervous system, participate in synaptic pruning, circuit refinement, and tissue homeostasis and can release cytokines under stress conditions [10,11,12,13,14,15]. Evidence for altered MG activity in depression is mixed: neuroimaging studies report increased translocator protein binding in subsets of patients, yet the interpretation of this signal and its specificity remain debated [16,17]. Similarly, elevations of peripheral cytokines (e.g., IL-6, TNF-α) have been observed in some cohorts; however, the extent to which peripheral signals cross the blood–brain barrier or reflect central immune activation is uncertain and it is likely less prominent than in neurodegenerative disease. Overall, inflammatory findings are heterogeneous across studies and patient subgroups, and causal directionality remains unresolved.

Antidepressants influence immune-related pathways in vitro and in vivo—including reductions in cytokine production and modulation of Toll-like receptor signaling—but these effects vary by compound, dose, exposure time, and biological context [18,19,20,21,22]. Thus, neuroimmune modulation is best viewed as a potential contributor to antidepressant action alongside effects on neurotransmission, synaptic plasticity, and cellular metabolism, rather than a universal or exclusive mechanism.

Lipids provide an additional layer linking membrane microdomains to receptor signaling and plasticity. Gangliosides, including GM3, organize membrane nanodomains and can modulate receptor-proximal signaling events [23,24,25,26,27,28]. ST3GAL5 encodes GM3 synthase, a glycosyltransferase that catalyzes GM3 biosynthesis and thereby has the potential to influence signaling pathways relevant to neuroinflammation and synaptic function [27,28,29,30]. Whether clinically used antidepressants converge on lipid remodeling pathways in microglia—and how such effects relate to inflammatory signaling—remains insufficiently defined.

To address these questions within a balanced framework of hypothesis generation, we profiled the transcriptomic responses of microglia to two approved antidepressants with distinct pharmacological properties, venlafaxine (VEN) and vortioxetine (VOR), and compared them with cells engineered to overexpress ST3GAL5 (ST3GAL5OE). We then performed functional validation using NF-κB and STAT3 reporter assays as well as cell viability measurements to determine whether the observed transcriptomic signatures translated into pathway-level functional changes.

## 2. Results

### 2.1. Selection of Clinically Relevant Concentrations for Transcriptomic Analyses

Pharmacokinetic considerations were taken into account when selecting concentrations of venlafaxine (VEN) and vortioxetine (VOR) for microarray experiments. Venlafaxine reaches steady state within approximately 4 days, with therapeutic plasma concentrations of the active moiety reported in the range of 140–600 ng/mL [31]. Vortioxetine achieves steady state within approximately 2 weeks, with mean C max values of 9–33 ng/mL after clinical dosing at 10–20 mg/day [32]. To ensure clinically relevant yet robust detection of transcriptional responses, we selected slightly higher but non-toxic concentrations: 200 ng/mL for VEN and 40 ng/mL for VOR. ATP viability assays confirmed that these concentrations were not cytotoxic to microglial cells (Appendix A). These concentrations were therefore applied throughout the transcriptomic profiling and subsequent functional experiments.

### 2.2. Transcriptomic Profiling Identifies a Convergent Gene Expression Signature Across VEN, VOR, and ST3GAL5 Overexpression

To explore the transcriptional impact of antidepressants and ST3GAL5 modulation, we conducted microarray profiling of microglial cells treated with VEN, VOR, or engineered to stably overexpress ST3GAL5 (ST3GAL5OE). After preprocessing, including log2(x + 1) transformation, Entrez ID updating, and duplicate symbol collapsing, a total of 19,769 unique genes were retained for downstream analysis. Using an exploratory threshold of |log2 fold change (FC)| ≥ 0.3 compared with control conditions, we identified a set of 41 genes consistently upregulated and 316 consistently downregulated across all three conditions (Figure 1).

Representative examples included APOE, which was robustly induced across all conditions (log2FC: VEN +0.70, VOR +0.45, ST3GAL5OE +0.35), consistent with a role in lipid remodeling. ST3GAL5 itself displayed reduced probe signal in microarray analysis, despite qRT-PCR validation confirming overexpression, reflecting a known probe-specific discrepancy. Other genes of interest included C3 (immune activation), HSPA5 (ER stress chaperone), DDIT3 (stress-induced transcription factor), and TREM2 (microglial homeostasis). A heatmap visualization highlighted the coordinated regulation of these and other genes, demonstrating convergence of VEN, VOR, and ST3GAL5OE on a shared transcriptional program (Appendix A).

Functional enrichment analyses provided mechanistic context. GO Biological Process enrichment of the 41 commonly upregulated genes revealed associations with synaptic remodeling and neuronal function. Representative categories visualized in Figure 2A include regulation of synapse organization, synaptic vesicle cycle, and neuron cell-cell adhesion, while the complete enrichment list is provided in Appendix A. Conversely, the 316 commonly downregulated genes were enriched for biological processes associated with reduced inflammatory mediator production. Representative GO BP categories are shown in Figure 2B, whereas the complete list is provided in Appendix A. KEGG pathway analysis further identified downregulated cascades such as Rap1, cAMP, Ras, MAPK, PI3K-Akt, and HIF-1 signaling (all adjusted *p* < 0.05), summarized in Appendix A.

Gene set enrichment analysis (GSEA) further supported these findings, highlighting coordinated regulation of synaptic and inflammatory pathways across conditions (Appendix A). Taken together, these transcriptomic results indicate that VEN, VOR, and ST3GAL5OE converge on a lipid remodeling and synaptic-supportive program while suppressing stress- and inflammation-related pathways in microglia.

### 2.3. Functional Validation of Cytokine-Induced Signaling Responses

To validate transcriptional predictions at the functional level, we employed microglial reporter lines stably expressing either NF-κB-RE-NlucP or SIE-NlucP to monitor activation of NF-κB and JAK/STAT3 signaling, respectively. In dose–response assays, TNF-α induced NF-κB activity in a concentration-dependent manner. Activation became detectable at ~0.03 ng/mL (~2-fold above baseline), increased to ~8-fold at 0.2 ng/mL, ~50-fold at 1 ng/mL, and reached ~180-fold at 100 ng/mL. Similarly, IL-6 induced STAT3 activity in a concentration-dependent manner: responses were negligible at ≤4 ng/mL (~1-fold), became detectable at ~20 ng/mL (~1.5-fold above baseline), and reached ~7-fold at 100 ng/mL (Figure 3A,B). Importantly, ATP viability assays demonstrated negligible toxicity across these cytokine titration ranges, as shown in Figure 3C,D for 24-h treatments. Longer 72-h incubations similarly showed only minimal effects (Appendix A), further supporting the validity of functional stimulation.

Real-time monitoring revealed distinct temporal activation profiles for the two cytokines (Appendix A). TNF-α induced rapid reporter activation, detectable within 2–3 h, peaking at ~6–8 h, and sustaining activity up to 24 h. In contrast, IL-6 elicited a gradual increase in reporter activity that continued to rise throughout the 24-h period, indicating a slower but sustained activation pattern. Collectively, these results validated the robustness of cytokine signaling in our microglial model system.

### 2.4. GM3 Suppresses Cytokine-Induced NF-κB and STAT3 Activity

Given that ST3GAL5 catalyzes GM3 biosynthesis, we next tested whether exogenous GM3 supplementation could modulate cytokine signaling. GM3 treatment alone (0–50 µM) did not induce additional cytotoxicity, as ATP assays showed that reductions in viability at higher concentrations were comparable to those observed with the DMSO vehicle (Appendix A). However, addition of 5 µM GM3 significantly suppressed cytokine-induced signaling, reducing TNF-α-induced NF-κB activity by ~26% at 10 ng/mL and IL-6-induced STAT3 activity by ~45% at 50 ng/mL compared with vehicle controls (Figure 4A,B). These effects were specific to signaling, as ATP levels remained unchanged (Figure 4C,D). These findings are consistent with the hypothesis that GM3 accumulation modifies membrane microdomains and dampens cytokine receptor signaling in microglia.

### 2.5. ST3GAL5 Overexpression Recapitulates GM3-Mediated Suppression

To directly assess the role of ST3GAL5, we generated microglial cells stably overexpressing ST3GAL5-DYK (ST3GAL5OE). Western blotting confirmed robust expression of the tagged enzyme (Figure 5A). Functionally, ST3GAL5OE cells exhibited attenuated cytokine responses compared with vector-transfected controls: TNF-α-induced NF-κB activity at 10 ng/mL was reduced by ~75%, and IL-6-induced STAT3 activity at 50 ng/mL was decreased by ~27% (Figure 5B,C). ATP viability was unaffected, indicating that reduced signaling was not due to general toxicity. These findings demonstrate that ST3GAL5 upregulation is sufficient to suppress microglial inflammatory signaling, consistent with transcriptomic predictions and GM3 supplementation assays.

### 2.6. Chronic Antidepressant Exposure Phenocopies ST3GAL5OE Effects

Finally, we examined whether chronic exposure to antidepressants could phenocopy the effects of ST3GAL5 overexpression. Microglial cells were cultured for >1 week in the presence of venlafaxine (200 ng/mL; VEN200) or vortioxetine (40 ng/mL; VOR40) before cytokine stimulation. Unlike ST3GAL5OE cells, the effects were modest and pathway-specific: TNF-α-induced NF-κB activity at 10 ng/mL was reduced by ~52% in VEN200 cells but remained largely unchanged in VOR40 cells, whereas IL-6-induced STAT3 activity at 50 ng/mL was reduced by ~16% in VOR40 cells but showed little change in VEN200 cells (Figure 6A,B). These functional results, although more modest, were consistent with the trends observed in ST3GAL5OE and GM3-treated cells, supporting the concept that antidepressants converge on the ST3GAL5–GM3 axis to suppress inflammatory NF-κB/STAT3 signaling in microglia.

## 3. Discussion

In this study, we integrated transcriptomic and functional approaches to explore how venlafaxine (VEN) and vortioxetine (VOR) regulate microglial activity. Microarray profiling revealed a convergent signature of 41 upregulated and 316 downregulated genes consistently shared across VEN, VOR, and ST3GAL5 overexpression (ST3GAL5OE) (Figure 1 and Appendix A). GO enrichment analyses highlighted synaptic organization and vesicle cycle pathways among the upregulated genes, while downregulated genes were enriched for nitric oxide biosynthesis and multiple pro-inflammatory cascades, including Rap1, cAMP, Ras, MAPK, and PI3K-Akt signaling (Figure 2; Appendix A). These transcriptomic patterns suggest that VEN and VOR, like ST3GAL5, drive a dual program of enhanced synaptic remodeling and reduced inflammatory signaling.

Functional assays provided direct evidence for this interpretation. Using microglial NF-κB and STAT3 reporter lines, we showed that TNF-α strongly and consistently activated inflammatory signaling, whereas IL-6 induced a more moderate response, both without affecting viability, consistent with transcriptomic predictions (Figure 3 and Appendix A). Importantly, both exogenous GM3 supplementation and ST3GAL5 overexpression attenuated cytokine-induced reporter activity without altering ATP levels (Figure 4, Figure 5 and Appendix A), directly linking the ST3GAL5–GM3 axis to suppression of inflammatory responses. Finally, chronic antidepressant exposure (VEN200, VOR40) phenocopied these effects (Figure 6), demonstrating that VEN and VOR converge functionally on lipid remodeling-mediated anti-inflammatory control.

The stronger suppression observed in ST3GAL5OE cells compared with chronic antidepressant treatment likely reflects direct versus indirect engagement of the GM3 pathway. Forced ST3GAL5 expression robustly increases GM3 synthesis, while venlafaxine and vortioxetine act more modestly, producing pathway-specific effects (VEN on NF-κB; VOR on STAT3). Moreover, GM3 exists as multiple molecular species with distinct ceramide chain compositions that differentially regulate immune and growth factor signaling. Thus, partial phenocopying by antidepressants may reflect selective shifts in GM3 species rather than a uniform increase in total GM3, highlighting the need for targeted lipidomic studies.

These results align with clinical observations that pro-inflammatory cytokines (e.g., IL-6, TNF-α) are elevated in major depression and decline after antidepressant treatment [33,34,35]. Within this context, our findings support a model in which VEN and VOR act, at least in part, through upregulation of ST3GAL5 and accumulation of GM3 ganglioside, leading to dampening of cytokine-specific signaling pathways. Prior work has shown that GM3 species modulate TLR4-dependent inflammatory responses in a lipid composition–dependent manner [28], consistent with our data implicating GM3 as an active mediator rather than a passive metabolite. Together, these findings expand the therapeutic framework of antidepressants beyond monoaminergic modulation to include lipid-mediated inflammatory regulation.

This study has some limitations. First, our study relied on immortalized human microglia lines, which cannot fully recapitulate the complexity of in vivo brain environments. Future studies in primary microglia or animal models will be required to validate these mechanisms under physiological conditions. Second, while ST3GAL5 probe signals appeared reduced in microarray data, qRT-PCR confirmed overexpression, indicating a technical limitation of probe annotation. Third, although GM3 effects were specific in our assays, experiments with other gangliosides as controls would strengthen mechanistic interpretation. Fourth, ATP assays are an indirect measure of cytotoxicity; additional methods (e.g., LDH release) should complement these results in future studies. Finally, while VEN and VOR consistently converged on the ST3GAL5–GM3 axis, generalization to all antidepressants should be made cautiously, and causal links to behavioral outcomes remain to be established.

From a translational perspective, our findings suggest that targeting lipid remodeling may provide novel therapeutic and biomarker strategies for MD. The pathway-specific actions of VEN (TNF-α/NF-κB suppression) and VOR (IL-6/STAT3 suppression) may help explain heterogeneity in clinical responses, particularly among patients with inflammation-driven pathophysiology. In this regard, GM3 and ST3GAL5 activity could serve as candidate biomarkers of antidepressant efficacy. Future clinical studies, including cerebrospinal fluid or postmortem brain analyses, will be essential to determine whether similar lipid remodeling mechanisms occur in patients with depression.

## 4. Materials and Methods

### 4.1. Reagents

Venlafaxine (VEN) hydrochloride (catalog no. V7264) and vortioxetine (VOR) hydrobromide (catalog no. SML3388) were purchased from Merck & Co., Inc. (Rahway, NJ, USA) and Sigma-Aldrich (MilliporeSigma, St. Louis, MO, USA), respectively. Human TNF-α recombinant protein (catalog no. 300-01A) and human IL-6 recombinant protein (catalog no. HZ-1019) were obtained from PeproTech, Inc. (Cranbury, NJ, USA) and Proteintech Group, Inc. (Rosemont, IL, USA), respectively. GM3 (ganglioside; catalog no. HY-114456) was purchased from MedChemExpress (Monmouth Junction, NJ, USA). X-tremeGENE™ 9 DNA Transfection Reagent (catalog no. XTG9-RO) was purchased from Roche Molecular Biochemicals (Penzberg, Germany). Puromycin (catalog no. ant-pr) and blasticidin S HCl (catalog no. A1113903) were purchased from Thermo Fisher Scientific Inc. (Waltham, MA, USA). XL-Bradford reagent was obtained from APRO Science Group (Kyoto, Japan).

VEN hydrochloride, human TNF-α, and human IL-6 were dissolved in phosphate-buffered saline (PBS), whereas VOR hydrobromide and GM3 were dissolved in dimethyl sulfoxide (DMSO). The final DMSO concentration in all experiments did not exceed 0.1% (*v*/*v*). Stock solutions were stored at –80 °C until use, except for X-tremeGENE™ 9 DNA Transfection Reagent, which was stored at 4 °C.

### 4.2. Cell Culture

An immortalized human microglia (MG) cell line (Applied Biological Materials Inc., Richmond, BC, Canada) was used. Cells were cultured in Prigrow III medium (Abcam, Cambridge, UK) supplemented with 10% heat-inactivated fetal bovine serum and 1% penicillin/streptomycin (Nacalai Tesque, Inc., Kyoto, Japan). Cells were maintained at 37 °C in a humidified atmosphere containing 5% CO_2_.

For stable reporter integration, a minimal promoter was ligated downstream of either the NF-κB-responsive element (NF-κB-RE) or the sis-inducible element (SIE) consensus sequence (Promega, Madison, WI, USA), followed by NanoLuc-PEST luciferase (Promega). The construct was cloned into the pCDH-Green-T2A-Puromycin vector (System Biosciences, Palo Alto, CA, USA). Lentiviruses were generated by co-transfecting HEK293TN cells (System Biosciences) with reporter plasmids and the pPACKH1 helper plasmid mix (pPACKH1-GAG, pPACKH1-REV, pVSV-G). MG cells were transduced and selected using puromycin. GFP fluorescence confirmed stable reporter integration.

For ST3GAL5 overexpression, a cDNA encoding human ST3GAL5 with a C-terminal DYKDDDDK tag was obtained from GenScript (Piscataway, NJ, USA) and cloned into a custom pCDH-Blasticidin vector. Lentiviruses were generated and used to transduce MG reporter cells. Stable integration was confirmed by blasticidin selection.

### 4.3. Cytotoxicity Assay

MG cells were seeded in 96-well white plates at a density of 1 × 10^3^ cells/well in 50 µL medium. After 24 h, test compounds were added at serial dilutions (2-fold or 3-fold) and incubated for 24–72 h. Cell viability was measured using the CellTiter-Glo^®^ Luminescent Cell Viability Assay (Promega). Luminescence was recorded with a plate reader, and viability was expressed relative to untreated controls (set to 1).

### 4.4. NanoLuc Reporter Assays

For endpoint analysis, MG reporter cells (NF-κB-RE-NlucP or SIE-NlucP) were seeded at 5 × 10^3^ cells/well in 96-well white plates. After 24 h, TNF-α or IL-6 was added at the indicated concentrations. After an additional 24 h incubation, Nluc activity was measured using the Nano-Glo^®^ Luciferase Assay System (Promega). Values were normalized to untreated controls (set to 1).

For real-time analysis, MG reporter cells were seeded at 1 × 10^4^ cells/well. Nano-Glo^®^ Endurazine™ Live Cell Substrate (Promega) was added simultaneously with TNF-α or IL-6, and luminescence was monitored manually every hour for the first 12 h and once at 24 h. All values were normalized to the 24 h time point.

For GM3 experiments, 5 µM GM3 was added at seeding and maintained for 24 h before cytokine stimulation. For ST3GAL5 overexpression (ST3GAL5OE), reporter assays were performed in parallel with vector controls. For chronic antidepressant pretreatment, MG cells were cultured with VEN (200 µM) or VOR (40 µM) for ≥7 days before cytokine stimulation.

### 4.5. RNA Extraction and Microarray

MG cells were treated with VEN (200 µM or 1 mM) or VOR (40 µM or 200 µM) for 24 h. Total RNA was purified using the miRNeasy Kit (Qiagen, Venlo, The Netherlands) and quantified with a NanoDrop 2000 (Thermo Fisher Scientific). RNA quality was assessed by Bioanalyzer 2100 (Agilent Technologies, Santa Clara, CA, USA).

DNA microarray was performed using the 3D-Gene^®^ Human Oligo Chip 25K v1.1 (Toray Industries, Tokyo, Japan), containing probes for 24,460 genes. Raw fluorescent signals were digitized and normalized per array (median intensity = 25). Background-subtracted values were log2(x + 1) transformed. Annotation was updated using org.Hs.eg.db3.21.0 (Bioconductor). Duplicated gene symbols were averaged; genes missing across all conditions were excluded; partially missing values were imputed as 0.

### 4.6. Differential Expression and Enrichment Analysis

Log2 fold-change (log2FC) values were calculated as:

VEN–Control,

VOR–Control,

ST3GAL5OE–Vector.

Genes with |log2FC| ≥ 0.3 were considered differentially expressed (exploratory threshold). Commonly up- and downregulated genes were identified across conditions.

GO Biological Process (BP) and KEGG pathway enrichment analyses were performed using clusterProfiler (v4.10.0). Adjusted *p* values < 0.05 (Benjamini–Hochberg correction) were considered significant. Gene set enrichment analysis (GSEA) was performed with gseGO and gseKEGG, using log2FC-ranked gene lists. Visualization employed enrichplot, ggplot2, and pheatmap.

### 4.7. Data Visualization and Statistics

Heatmaps were generated with pheatmap (row-scaled, correlation distance). Venn diagrams were created with ggvenn. PCA was performed using prcomp and visualized with ggplot2. Reporter and viability assays were conducted in triplicate (n = 3). Data are expressed as mean ± SD. Comparisons used Student’s *t*-test or one-way ANOVA with appropriate post hoc correction.

### 4.8. Western Blotting

ST3GAL5-DYK and control cells were lysed in buffer (50 mM Tris-HCl pH 8.0, 1 mM EDTA, 120 mM NaCl, 0.5% NP-40, 0.5 mM PMSF). Lysates were sonicated for 15 s and centrifuged (21,000× *g*, 10 min). Protein concentrations were determined using the XL-Bradford assay. Proteins (25 µg) were denatured in DTT, separated by SDS-PAGE (10% gel), and transferred to PVDF membranes. Membranes were probed with HRP-conjugated anti-DYK (1:10,000; Sigma-Aldrich) or anti-β-actin (1:10,000; Sigma-Aldrich), and detected by enhanced chemiluminescence. Band intensities were quantified using Multi Gauge v3.0 (Fujifilm, Tokyo, Japan).

## 5. Conclusions

This study demonstrates that venlafaxine (VEN) and vortioxetine (VOR) converge on a lipid remodeling pathway in microglia centered on ST3GAL5 and its product, the ganglioside GM3. These findings identify the ST3GAL5–GM3 axis as a mechanistic link between antidepressant treatment and microglial inflammatory regulation. From a translational perspective, our results extend the framework of antidepressant action beyond classical monoaminergic hypotheses, suggesting that lipid metabolism and ganglioside-mediated signaling contribute to therapeutic efficacy. Patients with heightened inflammatory profiles may particularly benefit from treatments that engage this pathway, and GM3 or ST3GAL5 activity—potentially assessed through lipidomic profiling of specific GM3 species—may hold promise as biomarkers for stratified treatment approaches.

## Figures and Tables

**Figure 1 ijms-26-09733-f001:**
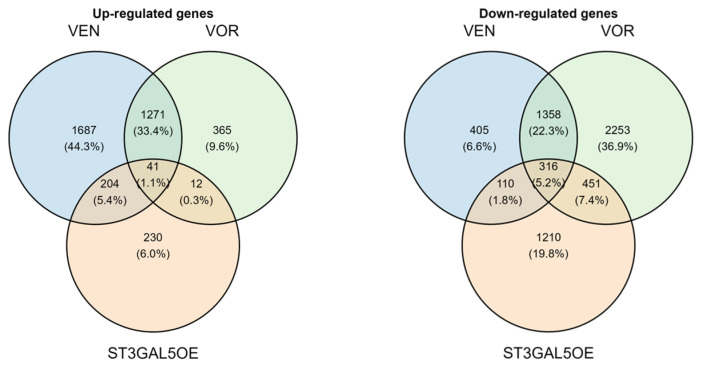
Commonly regulated genes across VEN, VOR, and ST3GAL5OE. Venn diagrams showing overlap of upregulated (**left**) and downregulated (**right**) genes (|log2FC| ≥ 0.3) shared among venlafaxine (VEN), vortioxetine (VOR), and ST3GAL5 overexpression (ST3GAL5OE). A total of 41 genes were commonly upregulated and 316 were commonly downregulated across all three conditions.

**Figure 2 ijms-26-09733-f002:**
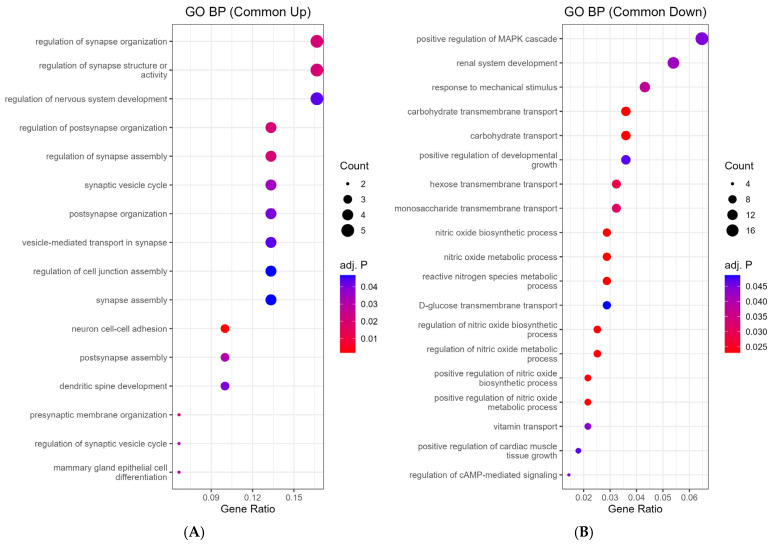
Functional enrichment of commonly regulated genes. (**A**) GO Biological Process enrichment of commonly upregulated genes, highlighting synapse-related pathways such as regulation of synapse organization, synaptic vesicle cycle, and neuron cell-cell adhesion. (**B**) GO Biological Process enrichment of commonly downregulated genes, including nitric oxide biosynthetic process and reactive nitrogen species metabolism, suggesting suppression of pro-inflammatory mediator production. Adjusted *p*-values are shown.

**Figure 3 ijms-26-09733-f003:**
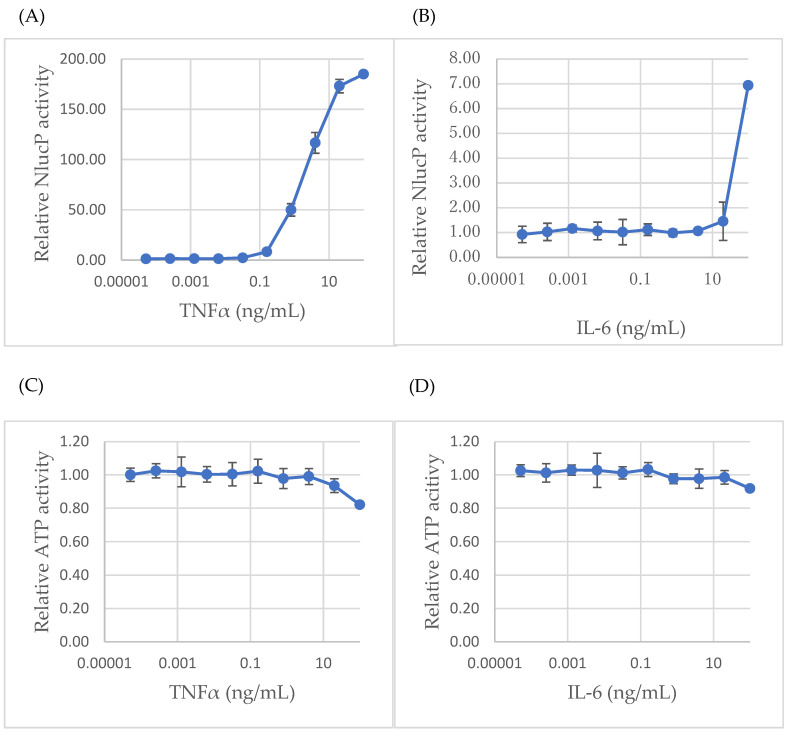
Reporter assays confirm cytokine-induced NF-κB and STAT3 activation. (**A**) Dose-response of NF-κB-RE-NlucP reporter cells stimulated with TNF-α (0–10 ng/mL). (**B**) Dose-response of SIE-NlucP reporter cells stimulated with IL-6 (0–50 ng/mL). (**C**,**D**) ATP viability assays showing minimal cytotoxicity under these conditions. Methods: MG cells stably expressing NF-κB-RE-NlucP or SIE-NlucP were seeded into 96-well white plates at 1 × 10^3^ cells/well in 50 µL medium. After 24 h, cells were stimulated with TNF-α (NF-κB-RE-NlucP) or IL-6 (SIE-NlucP) at the indicated concentrations (maximum 100 ng/mL for TNF-α; 50 ng/mL for IL-6) using 5-fold serial dilutions. Following 24 h incubation, NanoLuc activity was measured using the Nano-Glo^®^ Luciferase Assay System (**A**,**B**), and ATP levels were assessed with the CellTiter-Glo^®^ assay (**C**,**D**). Data represent mean ± SD (n = 3).

**Figure 4 ijms-26-09733-f004:**
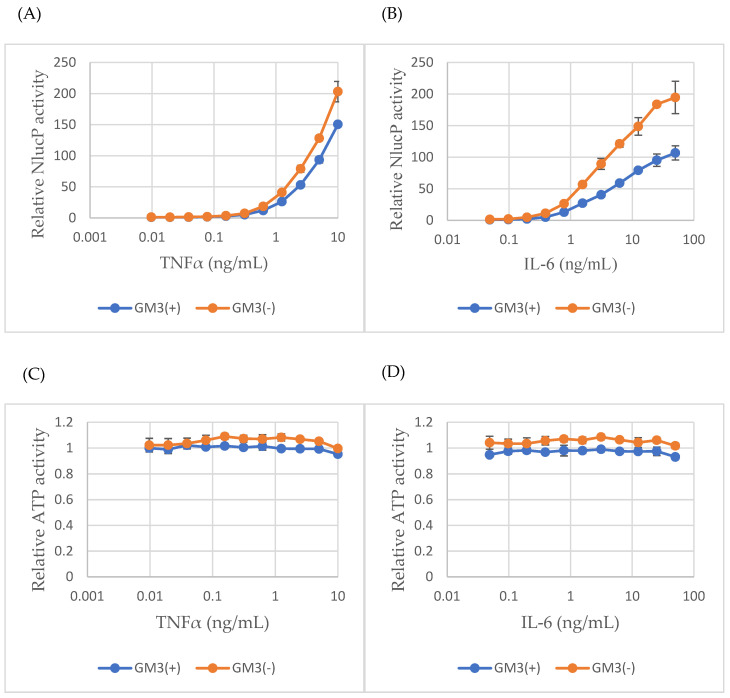
Exogenous GM3 suppresses cytokine-induced NF-κB and STAT3 activation. (**A**) GM3 (5 µM) reduced TNF-α-induced NF-κB activation. (**B**) GM3 reduced IL-6-induced STAT3 activation. (**C**,**D**) ATP viability assays confirmed no additional cytotoxicity with GM3 treatment. Methods: MG cells stably expressing NF-κB-RE-NlucP or SIE-NlucP were seeded into 96-well white plates at 5 × 10^3^ cells/well in 50 µL medium. GM3 (5 µM) was added at seeding. After 24 h, cells were stimulated with TNF-α (NF-κB-RE-NlucP) or IL-6 (SIE-NlucP) at the indicated concentrations (maximum 10 ng/mL for TNF-α; 50 ng/mL for IL-6) using 2-fold serial dilutions. Following a further 24 h incubation, NanoLuc activity was measured using the Nano-Glo^®^ Luciferase Assay System (**A**,**B**), and ATP levels were assessed with the CellTiter-Glo^®^ assay (**C**,**D**). Data represent mean ± SD (n = 3).

**Figure 5 ijms-26-09733-f005:**
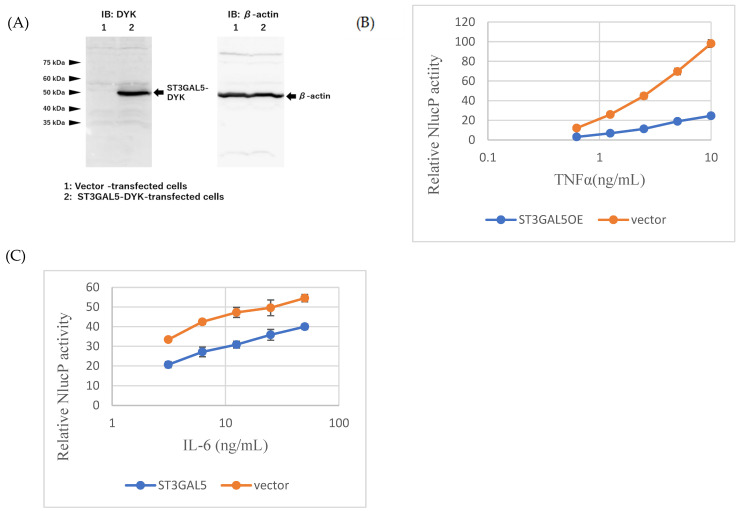
ST3GAL5 overexpression attenuates inflammatory signaling. (**A**) Western blot confirming ST3GAL5-DYK expression in transfected cells. Cell lysates from ST3GAL5-DYK stably expressing cells (ST3GAL5OE) were subjected to Western blotting, and the same membrane was sequentially probed with anti-DYK and anti-β-actin antibodies. (**B**,**C**) Reporter assays showing suppression of TNF-α-induced NF-κB and IL-6-induced STAT3 activity in ST3GAL5OE cells compared with vector controls. (**D**) ATP levels were unaffected, confirming no cytotoxicity. Methods: MG cells stably transfected with ST3GAL5-DYK (ST3GAL5OE) or vector control were seeded into 96-well white plates at 5 × 10^3^ cells/well in 50 µL medium. After 24 h, TNF-α (maximum 10 ng/mL) or IL-6 (maximum 50 ng/mL) was added with 2-fold serial dilutions. Following an additional 24 h incubation, NanoLuc activity was measured using the Nano-Glo^®^ Luciferase Assay System (**B**,**C**), Data represent mean ± SD (n = 3).

**Figure 6 ijms-26-09733-f006:**
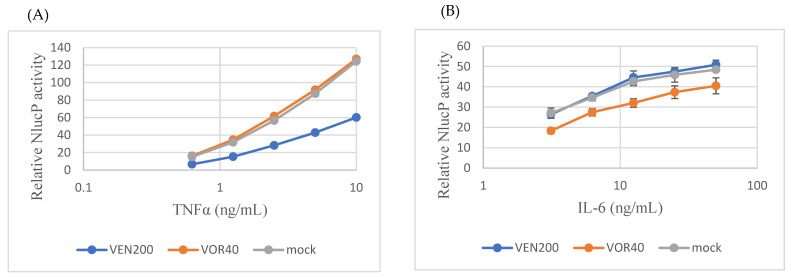
Chronic antidepressant exposure phenocopies ST3GAL5 effects. (**A**) VEN200 cells (cultured for >7 days in the presence of 200 µM venlafaxine) exhibited reduced TNF-α-induced NF-κB activation. (**B**) VOR40 cells (cultured for >7 days in the presence of 40 µM vortioxetine) exhibited reduced IL-6-induced STAT3 activation. Methods: MG cells were chronically pretreated with venlafaxine (200 µM) or vortioxetine (40 µM) for at least one week and designated as VEN200 and VOR40, respectively. These cells were seeded into 96-well white plates at 5 × 10^3^ cells/well in 50 µL medium. After 24 h, TNF-α (maximum 10 ng/mL, 2-fold serial dilutions) or IL-6 (maximum 50 ng/mL, 2-fold serial dilutions) was added. Following an additional 24 h incubation, NanoLuc activity was measured using the Nano-Glo^®^ Luciferase Assay System. Data are presented as mean ± SD (n = 3).

## Data Availability

The data presented in this study are available on request from the corresponding author.

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
