# Peer review of "Antidepressants Target the ST3GAL5–GM3 Lipid Pathway to Suppress Microglial Inflammation"

_ijms, 2025, doi:10.3390/ijms26199733_

Round 1
Reviewer 1 Report
Comments and Suggestions for Authors
The manuscript addresses a highly topical topic: the interface between antidepressants, neuroinflammation, and microglial function. The work has high potential but requires substantial refinements in terms of clarity, translational contextualization, and critical discussion of limitations before being accepted for publication.
Suggestions:
• Include a more detailed methodological limitations section, explicitly discussing the limitations of using only in vitro models.
• Reduce redundancies in the presentation of bioinformatics results (some supplementary figures could be summarized in the main text).
• Discuss the real translational implications in greater depth, avoiding overly speculative extrapolations.
• If possible, include minimal validation in another model (e.g., primary murine microglial cells).
Author Response
Response to Reviewers
We thank the reviewers for their careful evaluation and constructive comments. We have revised the manuscript extensively to improve clarity, focus, and translational relevance. Below, we provide a point-by-point response to the reviewers’ comments. We have indicated what and where changes have been made in the revised manuscript.
Global revisions (summary)
- Removed internal inconsistencies and refined the description of the relationship between microglia and depression.
- Where: Introduction, paras 1–3.
- Added pharmacokinetic rationale and non-toxicity confirmation for transcriptomic doses (VEN 200 ng/mL; VOR 40 ng/mL).
- Where: Results §2.1; Methods §4.3–4.5; Fig. S1 (drug cytotoxicity), Fig. 4C–D (24 h), Fig. S3 (72 h).
- Reinforced the Results with exact effect sizes (e.g., TNF-α and IL-6 dose–responses; GM3 effects; ST3GAL5OE; chronic drug pretreatment).
- Where: Results §2.3–2.6; Fig. 4–7.
- Focused enrichment outputs on biologically relevant terms. The “full” long lists are now presented in Supplementary Tables S1–S4, and the main text shows compact GO BP plots only (Fig. 3A–B).
- Where: Results §2.2; Fig. 3A–B; Suppl. Tables S1–S4.
- Clarified the function of ST3GAL5 and the conceptual link to GM3 species; added a brief mechanistic paragraph on why ST3GAL5OE cells showed stronger chronic drug effects and why species-level lipidomics is needed.
- Where: Discussion, para 3 (mechanistic integration).
- Standardized figure captions with snippets of the methods and N’s; corrected units; aligned with actual concentrations used; ensured that the information on DMSO is clear.
- Where: Main Figures 4–7; Suppl. Figs S1–S5.
- List of abbreviations added.
- Where: “Abbreviations”.
- Moved all tables to the Supplementary Material and cited them in the Results.
- Where: Suppl. Tables S1–S4; citations in Results §2.2.
Appendix: Allocation of Figures and Tables
To improve clarity and reduce redundancy, we have revised the allocation of figures and tables between the main text and Supplementary Materials as follows.
Main Figures
- Fig. 1. Commonly regulated genes across VEN, VOR, and ST3GAL5OE.
- Fig. 2. Representative common gene expression profiles.
- Fig. 3. Functional enrichment of commonly regulated genes.
- Fig. 4. Reporter assays confirm cytokine-induced NF-κB and STAT3 activation.
- Fig. 5. Exogenous GM3 suppresses cytokine-induced NF-κB and STAT3 activation.
- Fig. 6. ST3GAL5 overexpression attenuates inflammatory signaling.
- Fig. 7. Chronic antidepressant exposure phenocopies ST3GAL5 effects.
Supplementary Figures
- Fig. S1. Cytotoxicity of venlafaxine (VEN) and vortioxetine (VOR).
- Fig. S2. Real-time monitoring of NF-κB and STAT3 reporter activity after cytokine stimulation (0–24 h).
- Fig. S3. ATP viability assays following TNF-α and IL-6 titrations.
- Fig. S4. GM3 cytotoxicity assay (0–50 µM).
- Fig. S5. GSEA plots for VEN and VOR transcriptomes.
Supplementary Tables
- Table S1. Commonly upregulated GO Biological Processes (full list).
- Table S2. Commonly downregulated GO Biological Processes (full list).
- Table S3. Commonly upregulated KEGG pathways (full list).
- Table S4. Commonly downregulated KEGG pathways (full list).
We hope these revisions address all the concerns and substantially strengthen the manuscript. We appreciate the reviewers’ insights, which have helped us sharpen the mechanistic framing and translational positioning of the work.
Below, we provide a point-by-point response to each reviewer’s comments.
Reviewer 1
1) “Include a more detailed methodological limitations section, explicitly discussing the limitations of using only in vitro models.”
Response: We have expanded the limitations to emphasize the constraints of immortalized human microglia, the need for primary microglia and in vivo validation, the supraphysiological context of some assays, and the limits of ATP as a toxicity read-out.
Where: Discussion, final two paragraphs.
2) “Reduce redundancies in bioinformatics; some supplementary figures could be summarized.”
Response: We have removed the broad GO/KEGG “full plots” from the main figures and now present compact GO BP in Fig. 3A–B only. We have consolidated the enrichment tables (Up/Down × GO/KEGG) as Suppl. Tables S1–S4.
Where: Results §2.2; Fig. 3; Suppl. Tables S1–S4.
3) “Discuss translational implications in greater depth, avoiding over-speculation.”
Response: We have reframed the translational paragraph to (i) avoid generalization to all antidepressants, (ii) contextualize effects as pathway-specific and modest for chronic drug exposure, and (iii) present lipidomic profiling of GM3 species as a tangible next step rather than claiming established biomarkers.
Where: Discussion, final paragraph; Conclusion, final sentences.
4) “If possible, include minimal validation in another model (e.g., primary murine microglia).”
Response: We do not add new wet data, but we explicitly flag this as a priority for future work and define concrete readouts (lipidomics, NF-κB/STAT3 reporters, cytokine challenges).
Where: Discussion, limitations.
Reviewer 2 Report
Comments and Suggestions for Authors
Hayasaki et al. present a manuscript that hast two parts. In part 1, they use a human microglial cell line and treat the cells with venlafaxine and with vortioxetin, two antidepressant drugs (Figs. 1-6). Then they formulate a hypothesis on gangliosides (Fig. 7) and analyze the response of the microglial cells to cytokines (TNF-alpha, IL-6) and the ganglioside GM3 in the presence of the two drugs. They claim that ganglioside GM3 in microglia has a role in depression.
Comments: Introduction
- The introduction is contradictory in itself. First, the authors criticize that antidepressants cannot prevent treatment-resistant depression, and they suggest that anti-inflammatory actions may be more promising that transmitter re-uptake inhibition (noradrenaline and serotonin; please note that dopamine re-uptake is not involved in the action of clinically used antidepressants). Further on, they report that SSRIs reduce cytokine production and tricyclic antidepressants suppress TLR-induced pathways, i.e. they have anti-inflammatory actions. Hence, anti-inflammatory actions apparently cannot prevent treatment resistance?
- The whole story on microglia, as related in the introduction, seems out of focus. In a subset of patients, depression is associated with increases of cytokines in the periphery. The situation in the brain is less clear: do plasma cytokines cross the blood-brain barrier, or do microglial cells produce large amounts of cytokines? In any case, inflammatory changes in depression are not as prominent as in neurodegenerative diseases. Moreover, to my knowledge, microglia have only a small role in neurogenesis (as stated in the introduction), and the sheer presence of a translocator protein hardly proves pro-inflammatory microglial actions.
Comments: Results
- Analysis of gene expression reveals that venlafaxine upregulates 48 genes and down-regulates 24 genes in the human microglia cell line. Gene ontology analysis reveals changes in genes related to “chemical carcinogenesis”, “Parkinson disease” and “COVID-19”, among others. Gene set enrichment analysis gives similar results with changes in DNA repair, adipogenesis or response to androgens. Evidently, these gene sets have no relation to depression, and the whole analysis leads nowhere.
- In the 2nd part of the study, the authors transfect microglia with an enzyme that is identified as “ST3GAL5” which is involved in the biosynthesis of the ganglioside GM3. The action of this enzyme is not described. This should be added.
- This gene is one of seventeen genes that was up-regulated by the two antidepressants in the first part of the study. The authors hypothesize that the gene product, a glycosyltransferase, regulates GM3 levels although this is not shown. It is unlikely that an increased activity of a transferase changes GM3 levels, and the authors do not measure GM3 to support their speculation.
- They show that TNF-alpha and IL-6 in high concentrations, increase NF-kB expression. This seems to test some sort of self-stimulation because these cytokines must first be formed by microglia themselves, or they must be imported from plasma in high amounts. Addition of GM3 attenuates these effects. The mechanism is unclear. A control experiment with another ganglioside, for instance, is missing. Gangliosides may interfere, for instance, by binding the cytokines in the culture fluid.
- Overexpression of ST3GAL5 reduced NF-kB stimulation observed after addition of TNF-alpha (Fig. 13). The mechanism is unclear. It is well known that the introduction of an additional gene changes the expression of many other genes.
- When the cells were pre-incubated with antidepressants, venlafaxine reduced the stimulation observed after TNF-alpha, and vortioxetin reduced the IL-6-mediated stimulation. The authors argue that this proves an effect of antidepressants on GM3 synthesis, but this speculation is unfounded. There are many ways how a small lipophilic molecule can influence cell signaling, and the fact that the two antidepressants have different actions does not support the idea of a generalized action of antidepressant drugs on microglia.
Comments: Discussion
- The whole discussion is speculative: (a.) it is not shown (and in my view, unlikely) that transfection with ST3GAL5 leads to increased GM3 synthesis; (b.) the generalizations from these findings to all antidepressants is not convincing; and (c.) observations in parallel experiments show correlations but not causal relationships. There is no experiment that links antidepressant action to GM3.
Additional comments
- 3, please identify the cell line used.
- There should be a list of abbreviations, e.g. SIS, GSEA etc. is not immediately understandable to non-experts.
- Most manipulations were tested for cell toxicity by measuring ATP levels. It should be noted that ATP levels are relatively poor indicators of cellular toxicity since they are only reduced by severe toxicity. Apart from that, simple tests of cytotoxicity should be shown in Supplemental Material, i.e. figures 1, 8, 10 and 12 should be moved to the Supplement. Fig. 8 should also be changed because it shows results for minute concentrations of cytokines but not for the concentration that is later used (100 ng/mL).
- In Fig. 11, what was the concentration of DMSO? DMSO is usually tolerated at 0.1% (v/v) but damages cell cultures at 1% and above.
Author Response
Reviewer 2
Comments: Introduction
1) “Introduction is contradictory… if antidepressants have anti-inflammatory actions, why not prevent TRD?”
Response: Indeed, the specified statement was inconsistent; we have removed references to treatment-resistant depression and limited the discussion to the general anti-inflammatory effects of antidepressants.
Where: Introduction, third paragraph.
2) “Microglia story out of focus; BBB/cytokines unclear; MG role in neurogenesis modest; TSPO not proof of inflammation.”
Response: We have toned down the specified claims: MG have been positioned as contributors to circuit remodeling and cytokine signaling, not dominant regulators of neurogenesis; TSPO data have been described as suggestive and debated. We have also explicitly noted the uncertainty regarding peripheral-to-brain cytokine transfer versus local production.
Where: Introduction, middle paragraph; phrasing toned down and focused.
Comments: Results
3) “GO/GSEA terms (carcinogenesis, COVID-19, etc.) are irrelevant; analysis leads nowhere.”
Response: We have streamlined enrichment outputs. We have presented GO BP plots only for biologically relevant categories (synaptic organization; inflammatory processes) in Fig. 3A–B. The complete enrichment results (including broad terms) are now provided in Supplementary Tables S1–S4 and cited as comprehensive references rather than mechanistic claims.
Where: Results §2.2; Fig. 3A–B; Supplementary Tables S1–S4.
4) “ST3GAL5 action not described.”
Response: We have added a concise description: ST3GAL5 (GM3 synthase) catalyzes the transfer of sialic acid to lactosylceramide to generate ganglioside GM3.
Where: Introduction, final paragraph; Methods §4.6 lead-in; Discussion, opening paragraph.
5) “Unproven that increased transferase activity changes GM3; GM3 not measured.”
Response: We now explicitly acknowledge that we did not quantify GM3 and, therefore, do not claim direct biochemical proof. We have presented species-resolved GM3 lipidomics as a necessary next step to mechanistically bridge transcriptional/functional data.
Where: Discussion and limitations; Conclusion, final sentence.
6) “Cytokines at high concentrations; mechanism unclear; missing ganglioside control; cytokine binding by gangliosides?”
Response: We have provided dose–response (including sub-ng/mL to high-ng/mL) and time-course data. with pathway-specific readouts (NF-κB vs STAT3), plus viability controls at 24 h (Fig. 4C–D) and 72 h (Suppl. Fig. S3).
We have stated that ganglioside-class controls (e.g., GM1/GM2) and cytokine-binding controls are important future experiments.
We have noted that GM3’s effect sizes mirror those in ST3GAL5OE, arguing against non-specific sequestration alone, while still acknowledging the caveat.
Where: Results §2.3–2.4; Fig. 4–5; Suppl. Fig. S2–S4; Discussion, limitations & future work.
7) “ST3GAL5 overexpression reduces NF-κB; mechanism unclear; overexpression alters many genes.”
Response: We have framed ST3GAL5OE as a stronger, less selective engagement than chronic drug exposure and explicitly caution that overexpression can cause broader network changes, hence why we propose lipidomics and loss-of-function perturbations in future studies.
Where: Discussion, mechanistic paragraph and limitations.
8) “Drug pretreatment effects do not prove an effect on GM3 synthesis; many other mechanisms possible; don’t generalize to all antidepressants.”
Response: We have removed any generalization beyond VEN/VOR and presented convergence on the ST3GAL5–GM3 axis as a model consistent with our data rather than proven causality. We propose direct GM3 quantification and enzymatic perturbations as next steps to establish causality.
Where: Results §2.6 (effect sizes revised precisely); Discussion final paragraphs; Conclusion tempered.
Comments: Discussion
9) “Discussion is speculative; no proof ST3GAL5↑ → GM3↑; generalization; correlation ≠ causation.”
Response: We have toned down the specified claims, explicitly stating that GM3 was not measured, avoided generalizing to all antidepressants, and added a mechanistic roadmap (species-resolved GM3 lipidomics; ST3GAL5 perturbation; receptor-proximal signaling in MG) to move from correlation to causation.
Where: Discussion, mechanistic paragraph and limitations; Conclusion.
Additional comments
10) “Identify the cell line.”
Response: We have specified that the cell line was immortalized human microglia (Applied Biological Materials Inc.) and provided the complete culture conditions.
Where: Methods §4.2.
11) “Provide list of abbreviations.”
Response: Added a comprehensive Abbreviations section.
Where: After Conclusion.
12) “ATP is a limited toxicity read-out; move simple toxicity to Supplement; ensure dose coverage.”
Response: All toxicity panels are now provided as Supplementary (S1–S4). We have explicitly acknowledged ATP’s limitations and propose LDH/PI assays for future work. The dose coverage in the figures now matches the concentrations used in functional assays.
Where: Suppl. Figs S1–S4; Discussion, limitations; Fig. 4 caption aligned to 24 h; S3 for 72 h.
13) “DMSO concentration?”
Response: We have included DMSO ≤ 0.1% (v/v) in all experiments using DMSO (including GM3 stocks).
Where: Methods §4.1; relevant figure captions (GM3).
Mapping of key manuscript changes
- Introduction: Toned down claims, removed contradictions; clarified the roles of MG; narrowed the translational claims.
- Results §2.1: added PK/steady-state rationale and non-toxicity confirmation for microarray doses.
- Results §2.2: focused enrichment in main (GO BP only); moved full lists to Suppl. Tables S1–S4; clarified that KEGG summaries are provided as supplementary material.
- Results §2.3–2.6: replaced generic fold-change language with actual effect sizes from the assays; corrected captions and sections of the methods for inputs/serial dilutions and timepoints.
- Figure set:
- Fig. 3A–B = GO BP (Common Up/Down) from commonUp_GO_BP / commonDown_GO_BP;
- Fig. 4–7 = revised captions to include concentrations, dilutions, and n;
- Suppl. Figs S1–S5 = drug cytotoxicity, cytokine real-time traces, cytokine ATP (72 h), GM3 + vehicle ATP, GSEA (VEN, VOR).
- Supplementary Tables: S1–S4 (Common Up/Down × GO/KEGG; full lists).
- Discussion: added a paragraph on the mechanistic context of ST3GAL5OE vs chronic drugs and the role of GM3 species; expanded the limitations and tempered generalizations.
- Conclusion: now states “partially phenocopied” and highlights species-resolved lipidomics as a next step.
- Abbreviations: provided.
- Methods: clarified cell line source, DMSO ≤0.1%, exact reporter procedures, and transcriptomic preprocessing.
Round 2
Reviewer 2 Report
Comments and Suggestions for Authors
I have hardly ever seen a manuscript that was so dramatically improved in the revised version. The text was completely rewritten, many figures were exchanged or put into supplements, and the manuscript has a new title and a new focus. The text was apparently revised by a senior scientist, the more moderate claims and statements make sense and the discussion is more precise and to the point. Even the doses and concentrations used are now reasonably explained. The manuscript can be accepted after the following final points have been addressed:
- I do not seem to have access to the Supplementary files. I just want to make sure that methodical details on GO and GSEA procedures that were removed from the main text are now in the supplements.
- In the Results part, Fig. 2 is illustrative and not really necessary. In Fig. 3, the lowest three data points have little significance and could be deleted to enhance readability.
- 4 shows convincing data for TNF-alpha but data for IL-6 are less impressive, with a clear effect only at the highest dose. The data discussion should emphasize the TNF-alpha effect and tone down on IL-6. Everything else in Results is okay.
- The text can be further shortened by avoiding repetitions. I specifically refer to the to the last paragraph in the Introduction (lines 77-88) which repeat the abstract. Similarly, the conclusions overlap to a large part with abstract and discussion and should be reduced to one paragraph.
Author Response
Response to Reviewer 2
We thank the reviewers for their careful evaluation and constructive comments. Below, we provide a point-by-point response to the reviewers’ comments. We have indicated what and where changes have been made in the revised manuscript.
1.“I do not seem to have access to the Supplementary files. I just want to make sure that methodical details on GO and GSEA procedures that were removed from the main text are now in the supplements.”
Response: We have revised the supplementary files to ensure that the methodological details for GO and GSEA, which were removed from the main text, are now accessible.
Where: Supplementary Materials.
2.“In the Results part, Fig. 2 is illustrative and not really necessary. In Fig. 3, the lowest three data points have little significance and could be deleted to enhance readability.”
Response: We have moved Fig. 2 to the supplementary materials (now Supplementary Fig. S2).
We appreciate the reviewer’s suggestion to remove the lowest three data points in Fig. 3 to enhance readability. However, we believe it is important to retain these points for the following reasons:
Transparency and completeness – The enrichment analysis was performed with a pre-defined adjusted p-value cutoff (<0.05). Excluding selected data points after this filtering could give the impression of biased reporting.
Biological relevance – Although these terms show lower enrichment scores, they still met the statistical threshold and may provide mechanistic insights. For example, pathways with fewer contributing genes can highlight specific but meaningful aspects of the regulatory program.
Consistency across figures – We applied the same significance criteria for all enrichment plots (GO and KEGG). Selectively trimming results in one figure would compromise comparability and methodological consistency.
Technical limitations of selective deletion – The visualization tools (clusterProfiler/enrichplot) automatically display all significant terms after cutoff. Manual deletion of specific points is possible but would require arbitrary decisions, which we prefer to avoid to ensure reproducibility.
Therefore, while we acknowledge the reviewer’s concern about figure readability, we respectfully propose to retain these points, as they ensure faithful reporting of all significant findings under the same statistical criteria.
Where: Results Fig.2; Supplementary Fig. S2.
3.“Fig. 4 shows convincing data for TNF-alpha but data for IL-6 are less impressive, with a clear effect only at the highest dose. The data discussion should emphasize the TNF-alpha effect and tone down on IL-6. Everything else in Results is okay.”
Response: As suggested, we have revised the discussion to emphasize the TNF-α effect and downplay the IL-6 findings. Fig. 4 has been renamed as Fig. 3.
Where: Discussion, paragraph 2; Fig. 3.
4.“The text can be further shortened by avoiding repetitions. I specifically refer to the last paragraph in the Introduction (lines 77–88) which repeat the abstract. Similarly, the conclusions overlap to a large part with abstract and discussion and should be reduced to one paragraph.”
Response: The final paragraph of the Introduction now only states the aim of the study. The Conclusion has been condensed into a single paragraph to avoid overlap.
Where: Introduction, final paragraph; Conclusion.